# Equivalent Circuit of a Stacked Piezoelectric Cymbal Vibrator

**DOI:** 10.3390/mi15101205

**Published:** 2024-09-28

**Authors:** Zhaohan Gong, Yajun Zheng, Shuhan Yao, Xinhu Liu, Ningdong Hu, Hongping Hu

**Affiliations:** 1Department of Mechanics, School of Aerospace Engineering, Huazhong University of Science and Technology, Wuhan 430074, China; 2Hubei Key Laboratory for Engineering Structural Analysis and Safety Assessment, Huazhong University of Science and Technology, Wuhan 430074, China; 3Shanghai Ruisheng Kaitai Acoustic Science and Technology Co., Ltd., Shanghai 201100, China

**Keywords:** piezoelectric stack, cymbal, vibrator, equivalent circuit, finite element

## Abstract

In order to provide a convenient and fast calculation method, the equivalent circuit of a novel stacked piezoelectric cymbal vibrator is studied. The equivalent circuit model of the piezoelectric stack is derived by combining the equivalent circuit models of the thin piezoelectric disk and electrode. The equivalent circuit of the cymbal structure is then derived. The equivalent circuit model of the stacked piezoelectric cymbal vibrator is further proposed. The output axial displacements and output forces of the cymbal vibrator under different excitation voltages are investigated using the equivalent circuit model. The effectiveness of the equivalent circuit has been verified by comparing it with the finite element method. Furthermore, the equivalent circuit method has a much faster calculation speed than the finite element method.

## 1. Introduction

A piezoelectric cymbal transducer is a common piezoelectric energy conversion device, which is composed of a piezoelectric ceramic disk and two metal cymbals. It has the characteristics of low frequency, small size, large displacement, and high sensitivity. The working principle is that the piezoelectric ceramic disk generates radial vibration under external alternating current excitation. The metal cymbals convert the radial vibration into axial vibration. Moreover, the axial vibration amplitude is much greater than the radial vibration amplitude of the piezoelectric ceramic disk [1]. In practical applications, cymbal transducers are widely used in ultrasonic sensors, sonar systems, underwater communication, medical imaging, micro-electromechanical systems, and other fields [2,3,4,5]. However, due to the complex structure and working mechanism of cymbal transducer, it is still a challenge to make an accurate theoretical analysis of it. Numerous studies have been conducted on cymbal transducers since their emergence [6,7,8,9,10]. The finite element method is one of the main methods for studying cymbal transducers. For example, Li et al. [11] studied the longitudinal displacement of cymbal composite piezoelectric transducers and the relationship between longitudinal displacement and shape parameters of cymbals and piezoelectric ceramics. Lu et al. [12] calculated and analyzed the influence of structural parameters on output voltage and resonant frequency using the finite element method. Wu et al. [13] analyzed the geometric and physical factors affecting the axial displacement performance of the transducer. The above scholars conducted a systematic analysis of cymbal transducers using the finite element method. However, current research lacks a theoretical analysis of transducers, and the vibration characteristics of structures are still unclear.

The equivalent circuit method is an analytical method that equates the mechanical structure of vibration to a circuit. It can be used to analyze both the mechanical vibration of the structure and the coupling between the structure and the circuit. The calculation speed and efficiency can be further improved by simplifying the circuit. Because of its modularity, it can quickly identify and eliminate bugs. Furthermore, boundary conditions are easily imposed by changing circuit port values. Coupling between structures, and between structures and circuits, can be easily achieved through circuit port connections. In addition, it can help electrical engineers easily understand and analyze the characteristics of the circuit, optimize the performance of the circuit, and thus reduce design time and costs.

Many scholars have successfully studied the equivalent circuit models of transducers [14,15,16,17,18,19,20,21,22]. Shim has conducted a number of studies on cymbal transducers [23,24,25,26], but the equivalent circuit of the cone model has not yet been investigated for the analysis of metal cymbals. Therefore, it is necessary to establish an equivalent circuit model of the piezoelectric stack and cymbal coupling vibrator based on linear piezoelectric theory and thin shell theory.

Consequently, the equivalent circuit method (ECM) is applied to analyze the stacked piezoelectric cymbal vibrator. The output axial displacement and output force of the cymbal vibrator under different operating conditions are calculated using equivalent circuits. The correctness of the equivalent circuit is verified by comparing its results with those obtained by the finite element method (FEM). In the ECM, loss is introduced by replacing the material stiffness coefficient $c_{ij}^{E}$ with $c_{ij}^{E}\left(1+Qi\right)$, where *i*, *j* = 1, 2, 3, and i is the imaginary unit. The value of *Q* for ceramics is usually of the order of 10^−3^ to 10^−2^. Here, we have taken a relatively large value of *Q* = 1 × 10^−2^ because the device would likely encounter damping of other origins, e.g., the material’s electrical damping and air resistance, etc. [27,28]. In FEM analysis, the structural damping is set to 0.01.

## 2. Equivalent Circuit and Analysis of Piezoelectric Stack

As shown in Figure 1, the stacked piezoelectric cymbal vibrator consists of a piezoelectric stack and two cymbals. A circular piezoelectric stack is installed in the middle of the cymbal vibrator. The cymbals are shaped as cones. The base of the cymbals is bonded with a piezoelectric stack, while a small hole is introduced at the tip of the cymbals to maintain air pressure balance and reduce noise [29,30]. As shown in Figure 2, the circular piezoelectric stack is composed of alternating layers of piezoelectric material colored in yellow and electrodes, colored in blue. The polarization directions of the adjacent layers of the piezoelectric material are along the *z*-axis direction and the opposite direction. *h*_1_ and *h*_2_ respectively represent the thicknesses of each layer of the piezoelectric material and electrode. PZT-5H is chosen as the piezoelectric material, and silver is used as the electrode material. The materials’ parameters are listed in Table 1 [30]. Other material parameters are cited from the literature [31].

### 2.1. Equivalent Circuit Models of Piezoelectric Ceramic Disk and Electrode

#### 2.1.1. Piezoelectric Ceramic Disk

As an electromechanical coupling system, the cymbal vibrator can convert mechanical loading into output voltage, or voltage input into output displacement. To derive the equivalent circuit model of the cymbal vibrator, we first calculate the equivalent circuit models of the thin piezoelectric ceramic disk and electrode, respectively, then make up the piezoelectric stack. The detailed theoretical derivation can be found in [30,32,33,34].

For the cymbal vibrator, it satisfies *a* >> *h*_1_, where *a* denotes the radius of the thin piezoelectric ceramic disk. The second constitutive relationship of the piezoelectric material is given as follows:(1)$$\begin{array}{l} Τ=c^{E}S-e^{\prime }E\\ D=eS+\varepsilon^{S}E \end{array}$$
where **T** and **S** are stress and strain tensors, and **D** and **E** are the electric displacement vector and electric field vector, respectively. $c^{E}$ is the stiffness coefficient matrix under a constant electric field, **e**′ is transpose of the piezoelectric stress constant matrix, and $\varepsilon^{S}$ is the clamping dielectric constant. 

As a thin plate, the piezoelectric ceramic sheet can be assumed to be *T_zz_* = 0 [35]. The motion equation and strain-displacement relationship yield the following:(2)$$\frac{\partial u_{z}}{\partial z}=-\left(\frac{c_{13}^{E}}{c_{33}^{E}}\frac{\partial u_{r1}}{\partial r}+\frac{c_{13}^{E}}{c_{33}^{E}}\frac{u_{r1}}{r}+\frac{e_{33}}{c_{33}^{E}}\frac{\partial \phi }{\partial z}\right)$$
(3)$$\begin{array}{l} T_{rr}=c_{11}^{p}\frac{\partial u_{r1}}{\partial r}+c_{12}^{p}\frac{u_{r1}}{r}+e_{31}^{p}\frac{\partial \phi }{\partial z}\\ T_{\theta \theta }=c_{12}^{p}\frac{\partial u_{r1}}{\partial r}+c_{11}^{p}\frac{u_{r1}}{r}+e_{31}^{p}\frac{\partial \phi }{\partial z}\\ T_{rz}=c_{44}^{E}\left(\frac{\partial u_{r1}}{\partial r}+\frac{\partial u_{z1}}{\partial r}\right)\\ D_{z}=e_{31}^{p}\frac{\partial u_{r1}}{\partial r}+e_{31}^{p}\frac{u_{r1}}{r}-\varepsilon_{33}^{p}\frac{\partial \phi }{\partial z} \end{array}$$
where
(4)$$\frac{\partial \phi }{\partial z}=-\frac{V}{h_{1}}$$
and
(5)$$\begin{array}{l} c_{11}^{p}=c_{11}^{E}-{\left(c_{13}^{E}\right)}^{2}/c_{33}^{E},\;c_{12}^{p}=c_{12}^{E}-{\left(c_{13}^{E}\right)}^{2}/c_{33}^{E}\\ e_{31}^{p}=e_{31}-c_{13}^{E}e_{33}/c_{33}^{E},\;\varepsilon_{33}^{p}=\varepsilon_{33}^{S}-{\left(e_{33}\right)}^{2}/c_{33}^{E}, \end{array}$$
$u_{r1}$ and $u_{z1}$ are the radial and axial displacements of the piezoelectric ceramic disk, respectively; $T_{rr}$, $T_{\theta \theta }$, and $T_{rz}$ are the radial normal stress, the tangential normal stress, and the shear stress, respectively. $D_{z}$ is the electric displacement. The solution of the radial displacement is written as follows:(6)$$u_{r1}\left(r\right)=B_{1}J_{1}\left(\xi_{1}r\right)e^{i\omega t}$$
where *B*_1_ is an undetermined constant, *ω* is the frequency, i is the imaginary unit, and $J_{1}\left(\xi_{1}a\right)$ is the first-order Bessel function, where $\xi_{1}=\omega \sqrt{\rho_{1}/c_{11}^{p}}$.

For the equivalent circuit method, the undetermined constant is always given by a boundary value:(7)$$B_{1}=\frac{v_{a1}}{i\omega J_{1}\left(\xi a\right)}$$
where $v_{a1}$ is the radial velocity, with $v_{a1}=i\omega u_{r1}\left(a\right)$.

From Equation (3), the radial tension force, at *r* = *a*, of the piezoelectric ceramic disk is
(8)$$F_{1}=-S_{1}T_{rr}=\frac{S_{1}c_{11}^{p}\xi_{1}i}{\omega }\left[-\frac{J_{0}\left(\xi_{1}a\right)}{J_{1}\left(\xi_{1}a\right)}+\frac{c_{11}^{p}-c_{12}^{p}}{c_{11}^{p}\xi_{1}a}\right]v_{a1}+nV$$
where the voltage ratio of the transformer is $n=2\pi ae_{31}^{p}$, the piezoelectric layer side area is $S_{1}=2\pi ah_{1}$, and $\rho_{1}$ is the density of the piezoelectric material.

From Equation (3), the total charge on the bottom surface of the thin disk is obtained as follows:(9)$$Q=\oint_{A}D_{z}dA=\frac{\pi a^{2}\varepsilon_{33}^{p}}{h_{1}}V-2\pi ae_{31}^{p}B_{1}J_{1}\left(\xi a\right)$$

From the relationship between the current and charge, I=dQ/dt, we have I=iωQ. From Equations (4) and (7)–(9), the circuit state equation is given by
(10)$$I=i\omega C_{0}V-nv_{a1}$$
where the static capacitance of the thin piezoelectric disk is $C_{0}=\pi a^{2}\varepsilon_{33}^{p}/h_{1}$

From Equation (8), we have
(11)$$F_{1}=Z_{1}v_{a1}+nV$$
where
(12)$$Z_{1}=\frac{S_{1}c_{11}^{p}\xi i}{\omega }\left[-\frac{J_{0}\left(\xi a\right)}{J_{1}\left(\xi a\right)}+\frac{c_{11}^{p}-c_{12}^{p}}{c_{11}^{p}\xi a}\right]$$

Based on Equations (10) and (11), the equivalent circuit of the thin piezoelectric ceramic disk in radial vibration is plotted in Figure 3.

#### 2.1.2. Electrodes

As the isotropic elastic disk, silver electrodes vibrate axisymmetrically in a radial direction.
(13)$$\frac{\partial u_{z2}}{\partial z}=-\frac{\lambda }{\lambda +2G}\left(\frac{\partial u_{r2}}{\partial r}+\frac{u_{r2}}{r}\right)$$
(14)$$\begin{array}{l} \sigma_{r2}=\lambda_{1}^{p}\frac{\partial u_{r2}}{\partial r}+\lambda_{2}^{p}\frac{u_{r2}}{r}\\ \sigma_{\theta 2}=\lambda_{2}^{p}\frac{\partial u_{r2}}{\partial r}+\lambda_{1}^{p}\frac{u_{r2}}{r}\\ \tau_{zr2}=G\frac{\partial u_{r2}}{\partial z} \end{array}$$
where $u_{r2}$ and $u_{z2}$ are the radial and axial displacements of the electrode, respectively. *λ* and *G* are the Lame’s constant and shear modulus. $\sigma_{r2}$, $\sigma_{\theta 2}$, and $\tau_{zr2}$ are the radial normal stress, circumferential normal stress, and shear stress, respectively.
(15)$$\lambda_{1}^{p}=\lambda +2G-\frac{\lambda^{2}}{\lambda +2G},\lambda_{2}^{p}=\lambda -\frac{\lambda^{2}}{\lambda +2G}$$

Similarly, expressions of $u_{r2}$ and $F_{2}$ are derived as follows:(16)$$u_{r2}=B_{2}J_{1}\left(\xi_{2}r\right)e^{i\omega t}$$
where *B*_2_ is an undetermined constant, $\xi_{2}=\omega \sqrt{\rho_{2}/\lambda_{1}^{p}}$, and $\rho_{2}$ is the density of silver.
(17)$$F_{2}=Z_{2}v_{a2}$$
where $F_{2}$ is the radial force on the silver electrode, and $u_{r2}$ is the radial vibration velocity of the silver electrode.
(18)$$Z_{2}=\frac{S_{2}}{i\omega }\left[\frac{\lambda_{1}^{p}\xi_{2}J_{0}\left(\xi_{2}a\right)}{J_{1}\left(\xi_{2}a\right)}+\frac{2G}{a}\right]$$
where $S_{2}=2\pi ah_{2}$. The equivalent circuit of the circular electrode in radial vibration is then plotted in Figure 4.

### 2.2. Comparison between Equivalent Circuit and Finite Element Method of Piezoelectric Stack

#### 2.2.1. Equivalent Circuit of Piezoelectric Stack

The piezoelectric stack is composed of *N* layers of thin piezoelectric disks and *N* + 1 layers of electrodes. The radial force *F_N_* of the piezoelectric stack is written as follows:(19)$$\begin{array}{l} F_{N}=NF_{1}+\left(N+1\right)F_{2}\\ =N\left(Z_{1}v_{aN}+nV\right)+\left(N+1\right)Z_{2}v_{aN}\\ =\left[NZ_{1}+\left(N+1\right)Z_{2}\right]v_{aN}+nNV \end{array}$$
where $v_{aN}$ is the radial vibration velocity of the piezoelectric stack.

The circuit state equation is written as follows:(20)$$I_{N}=NI=i\omega NC_{0}V-nNv_{aN}$$

Based on Equations (19) and (20), the equivalent circuit of the piezoelectric stack is shown in Figure 5.

#### 2.2.2. Comparing Results of Two Methods

A numerical model is calculated by the finite element method (FEM) in Comsol Multiphysics 6.1. The piezoelectric stack with *N* = 19 layers is excited by an alternating voltage with an amplitude of 80 V and a frequency of 100 Hz. The boundaries of the piezoelectric stack are assumed as free, i.e., *F_N_* = 0.

From Figure 6, one can observe that the maximum value of the radial displacement is 2.82 µm. The maximum value of the radial displacement is further calculated by the equivalent circuit method (ECM). Here, the same boundary conditions, structure, and sizes are adopted as those in Figure 6, with *N* = 19 and a frequency of 100 Hz.

Figure 7 illustrates the maximum radial displacements versus driving voltages obtained by the finite element method (FEM) and ECM. For the results of the FEM, the maximum point is marked on the line with its coordinates (80 V, 2.82 µm). One can observe that the maximum radial displacement obtained by the ECM is consistent with that achieved by the FEM.

## 3. Equivalent Circuit Model of Cymbal Disk and Comparison between Two Methods

As shown in Figure 1, the metal cap connected with the piezoelectric stack can be regarded as a cymbal-shaped disk, which consists of a tip plane with a hole, a conical shell on the side, and a base that acts as a paste area, as shown in Figure 8. The cymbal disk can convert the small radial displacement of the piezoelectric stack into a large axial displacement. Since the displacement amplification effect is mainly generated by the side conical shell, the cymbal-shaped structure can be approximately considered as a conical thin shell, as shown in Figure 9. The relevant parameters of the material are shown in Table 2.

### 3.1. Equivalent Circuit Model of Cymbal Disk

A cone-shaped axisymmetric model is proposed for the cymbal disk. The relevant parameters of the equivalent circuit can be derived by using the transfer matrix method [15,30,36,37,38,39]. The hypotenuse of the cone is divided into *m* segments. Because of axial symmetry, we have
(21)$$v=0,\;\frac{\partial }{\partial \theta }=0$$

The displacements and internal forces of *i*-th segment are expressed as follows:(22)$$\left[\begin{array}{c} u\\ w\\ \varphi \\ N_{s}\\ M_{s}\\ Q_{s} \end{array}\right]=\left[\begin{array}{cccccc} e^{\lambda_{1}^{i}s} & e^{\lambda_{2}^{i}s} & e^{\lambda_{3}^{i}s} & e^{\lambda_{4}^{i}s} & e^{\lambda_{5}^{i}s} & e^{\lambda_{6}^{i}s}\\ q_{1}^{i}e^{\lambda_{1}^{i}s} & q_{2}^{i}e^{\lambda_{2}^{i}s} & q_{3}^{i}e^{\lambda_{3}^{i}s} & q_{4}^{i}e^{\lambda_{4}^{i}s} & q_{5}^{i}e^{\lambda_{5}^{i}s} & q_{6}^{i}e^{\lambda_{6}^{i}s}\\ q_{1}^{i}\lambda_{1}^{i}e^{\lambda_{1}^{i}s} & q_{2}^{i}\lambda_{2}^{i}e^{\lambda_{2}^{i}s} & q_{3}^{i}\lambda_{3}^{i}e^{\lambda_{3}^{i}s} & q_{4}^{i}\lambda_{4}^{i}e^{\lambda_{4}^{i}s} & q_{5}^{i}\lambda_{5}^{i}e^{\lambda_{5}^{i}s} & q_{6}^{i}\lambda_{6}^{i}e^{\lambda_{6}^{i}s}\\ N_{s1}^{i} & N_{s2}^{i} & N_{s3}^{i} & N_{s4}^{i} & N_{s5}^{i} & N_{s6}^{i}\\ M_{s1}^{i} & M_{s2}^{i} & M_{s3}^{i} & M_{s4}^{i} & M_{s5}^{i} & M_{s6}^{i}\\ Q_{s1}^{i} & Q_{s2}^{i} & Q_{s3}^{i} & Q_{s4}^{i} & Q_{s5}^{i} & Q_{s6}^{i} \end{array}\right]\left[\begin{array}{c} A_{1}^{i}\\ A_{2}^{i}\\ A_{3}^{i}\\ A_{4}^{i}\\ A_{5}^{i}\\ A_{6}^{i} \end{array}\right]$$
where *u* and *w* are displacements in the *s* direction and its perpendicular direction, respectively, and $N_{s}$, $M_{s}$, and $Q_{s}$ are the normal forces, bending moments, and shear force in the *s* direction.
(23)$$\begin{array}{l} q_{j}\left(\lambda \right)=-\left[\left(\frac{\rho h\omega^{2}}{D}+\lambda_{j}^{2}\right)s^{2}+\lambda_{j}s-1\right]\left(\frac{\tan\alpha }{\mu \lambda_{j}s-1}\right)\\ \varphi =\partial w/\partial s\\ N_{sj}^{i}=D\left[\lambda_{j}^{i}e^{\lambda_{j}^{i}s}+\frac{\mu }{s}\left(e^{\lambda_{j}^{i}s}+\frac{q_{j}^{i}e^{\lambda_{j}^{i}s}}{\tan\alpha }\right)\right]\\ M_{sj}^{i}=K\left(\lambda_{j}^{i}+\frac{\mu }{s}\right)\lambda_{j}^{i}q_{j}^{i}e^{\lambda_{j}^{i}s}\\ Q_{sj}^{i}=K\left(\lambda_{j}^{i2}+\frac{1}{s}\lambda_{j}^{i}-\frac{1}{s^{2}}\right)\lambda_{j}^{i}q_{j}^{i}e^{\lambda_{j}^{i}s} \end{array}$$
where $D=Eh_{0}/\left(1-\mu^{2}\right)$ and $K=Eh_{0}^{3}/\left[12\left(1-\mu^{2}\right)\right]$. The roots of $\lambda_{j}$ are obtained by the following equation of the determinant:(24)$$\left|\begin{array}{cc} \frac{\mu s_{j}\lambda_{j}-1}{\tan\alpha } & \left(\frac{\rho h\omega^{2}}{D}+\lambda_{j}^{2}\right)s_{j}^{2}+s_{j}\lambda_{j}-1\\ Ks_{j}\lambda_{j}\left(s_{j}^{3}\lambda_{j}^{3}+2s_{j}^{2}\lambda_{j}^{2}-s_{j}\lambda_{j}+1\right)+\frac{Ds_{j}^{2}}{\tan^{2}\alpha }-s_{j}^{4}\rho h\omega^{2} & \frac{\mu s_{j}^{2}\left(1+\mu s_{j}\lambda_{j}\right)}{\tan\alpha } \end{array}\right|=0$$
where the coordinate *s_j_* is the coordinate of the midpoint of the *j*-th segment.

The radial force and displacement on the cymbal base (*s* = *L*) are obtained by the balance of forces, where L=h/cosα.
(25)$$F_{3}=N_{s}\left(L\right)\sin\alpha +Q_{s}\left(L\right)\cos\alpha$$
(26)$$u_{r}\left(a\right)=u\left(L\right)\sin\alpha +w\left(L\right)\cos\alpha$$

The impedance of the cymbal disk in radial vibration can be calculated as follows:(27)$$\begin{array}{l} Z_{3}=\frac{F_{3}}{i\omega \;u_{r}\left(a\right)}\\ =\left\{\sin\alpha \sum_{j=1}^{6}D\left[\lambda_{j}^{m}e^{\lambda_{j}^{m}L}+\frac{\mu }{L}\left(e^{\lambda_{j}^{i}L}+\frac{q_{j}^{m}e^{\lambda_{j}^{m}L}}{\tan\alpha }\right)\right]A_{j}^{m}\right.\\ \left.+\cos\alpha \sum_{j=1}^{6}K\left(q_{j}^{m}\lambda_{j}^{m3}e^{\lambda_{j}^{m}L}+\frac{1}{L}q_{j}^{m}\lambda_{j}^{m2}e^{\lambda_{j}^{m}L}-\frac{1}{L^{2}}q_{j}^{m}\lambda_{j}^{m}e^{\lambda_{j}^{m}L}\right)A_{j}^{m}\right\}\\ /\left\{i\omega \left[\sin\alpha \sum_{j=1}^{6}e^{\lambda_{j}^{m}L}A_{j}^{m}+\cos\alpha \sum_{j=1}^{6}q_{j}^{m}e^{\lambda_{j}^{m}L}A_{j}^{m}\right]\right\} \end{array}$$

The equivalent circuit model of the cymbal is shown in Figure 10. $v_{a3}$ is the radial vibration velocity of the cymbal base. *v_z_* and *F_z_* are the axial velocity and force on the tip of the cymbal. *T* represents the amplification factor for converting radial vibration speed to axial vibration speed, which is written as follows:(28)$$T=\frac{-\sin\alpha \sum_{j=1}^{6}A_{j}^{m}+\cos\alpha \sum_{j=1}^{6}q_{j}^{m}A_{j}^{m}}{\sin\alpha \sum_{j=1}^{6}e^{\lambda_{1}^{n}L}A_{j}^{m}+\cos\alpha \sum_{j=1}^{6}q_{j}^{m}e^{\lambda_{1}^{n}L}A_{j}^{m}}$$

### 3.2. Comparison Results of Cymbal between Equivalent Circuit and Finite Element Methods

Finite element simulation was carried out in the software Comsol 6.1. Firstly, the three-dimensional cymbal model was established. For the cymbal, the tip is free, i.e., *F_z_* = 0, and the base is driven by a radial displacement.

The axial displacement *u_z_* of the tip versus the radial displacement *u_r_* of the base was calculated by FEM and ECM, respectively. The results are compared in Figure 11. It can be noted that the radial displacement is converted into axial displacement by the cymbal, and the value is amplified by nearly 15 times. Furthermore, the results obtained by the ECM agree well with those calculated by the FEM.

## 4. Overall Analysis of Stacked Cymbal Vibrator

### 4.1. Equivalent Circuit Model of Stacked Cymbal Vibrator

Consider the overall structure of the cymbal vibrator. The radial force is balanced between the piezoelectric stack and the cymbal.
(29)$$F_{N}\left(a\right)-2\left[N_{s}\left(L\right)\sin\alpha +Q_{s}\left(L\right)\cos\alpha \right]=0$$

With free boundaries, the relation between the radial vibration speed *v_r_* and the excitation voltage *V* can be deduced from Equation (19) to Equation (29).
(30)$$v_{r}=-\frac{nNV}{NZ_{1}+\left(N+1\right)Z_{2}-2Z_{3}}$$

An equivalent circuit of the whole system is shown in Figure 12. The axial force *F_zi_*, where *i* = 1 and 2, denotes the lower and upper cymbals, respectively. On the cymbal tip, the axial force becomes
(31)$$F_{zi}=Q_{s}\sin\alpha -N_{s}\cos\alpha$$

If the tip of the lower cymbal is fixed, then *v_z_*_1_ = 0. However, the cymbal at the bottom still has an amplifying effect. The equivalent circuit can further be simplified to Figure 13. The total impedance becomes
(32)$$Z=NZ_{1}+(N+1)Z_{2}+2Z_{3}$$

### 4.2. Comparison between Results of Finite Element and Equivalent Circuit

As shown in Figure 14, a model of the overall structure was established in SolidWorks, which was then imported into Comsol for calculation. The tip of the lower cymbal is fixed. A steady-state analysis was conducted to obtain the output axial displacement and force under different voltage excitations. Firstly, consider the tip of the upper cymbal as free. Figure 15 shows the cloud diagram of the axial displacement under an 80 V voltage. The maximum axial displacement is 77.7 µm, which appears at the tip of the upper cymbal.

An axial harmonic compression force with frequency 100 Hz is applied on the top of the vibrator along the *z* direction. Figure 16 demonstrates the output axial displacement versus the excitation voltage under different output forces. The difference between the results of the ECM and the FEM is less than 5%. Furthermore, on the same computer, the calculation time for solving each output force for the FEM is about 20 min and that for ECM is about 10 s.

Figure 17 illustrates the axial displacement versus operating frequency under different voltages. The operating frequency of the device is less than 1000 Hz, far below the resonant frequency of its structure. Similarly, compared with the FEM, the ECM has a faster calculation speed. Moreover, the axial displacement does not change significantly with frequency under the same excitation voltage. This is because the driving frequency is far from the resonance frequency.

## 5. Conclusions

The equivalent circuit model of the new stacked piezoelectric cymbal vibrator has been proposed to investigate its performance under different electrical and mechanical boundary conditions. The equivalent circuits of the piezoelectric stack and the cymbals were established first, and then the equivalent circuit of the stacked cymbal vibrator was combined as a whole. By comparing the results of the proposed model and those of the finite element method, the effectiveness of the equivalent circuit model was further verified. Furthermore, the calculation speed of the equivalent circuit model is two orders of magnitude faster than that of the three-dimensional finite element model. Therefore, the equivalent circuit method not only simplifies the calculation process, but also provides a simple and effective method for predicting and optimizing the performance of the piezoelectric device.

## Figures and Tables

**Figure 1 micromachines-15-01205-f001:**
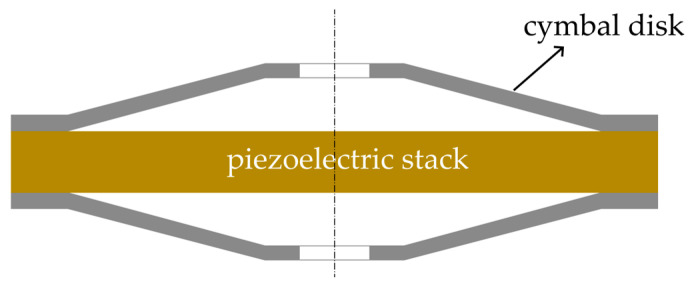
Cross-section of the cymbal vibrator [30].

**Figure 2 micromachines-15-01205-f002:**
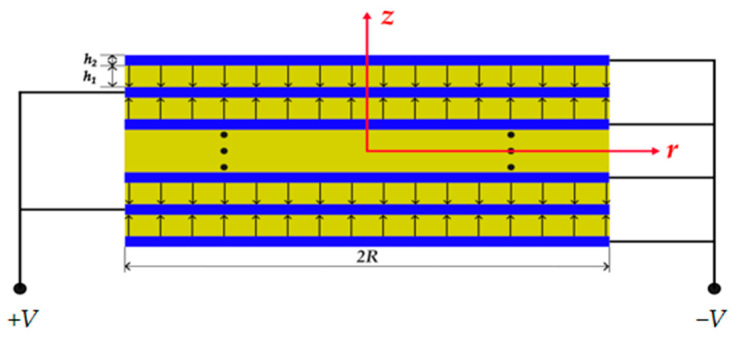
Cross-section of the piezoelectric stack [30]. Black dots indicate structural duplication.

**Figure 3 micromachines-15-01205-f003:**
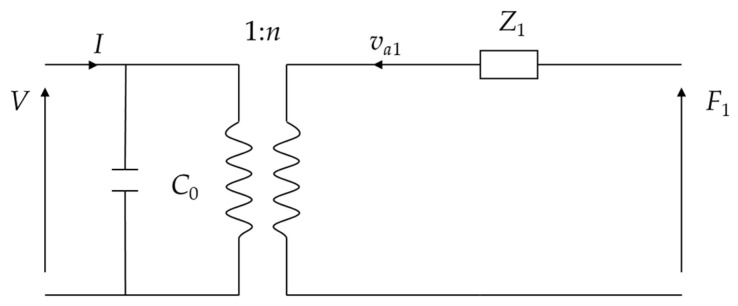
Equivalent circuit of a thin piezoelectric disk in radial vibration.

**Figure 4 micromachines-15-01205-f004:**
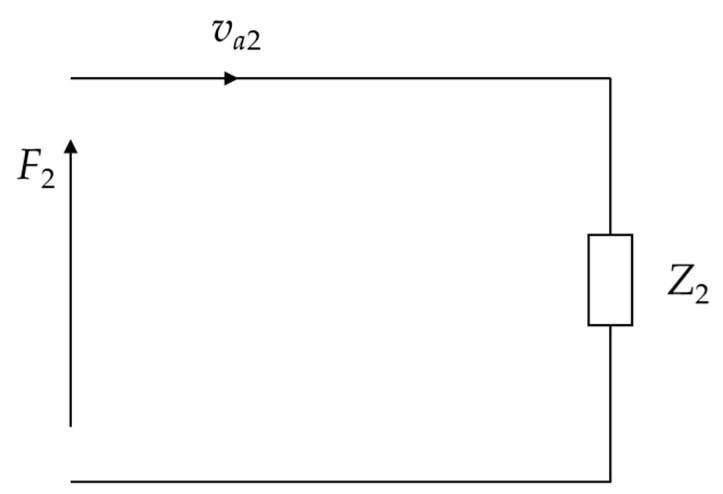
Equivalent circuit of the circular electrode in radial vibration.

**Figure 5 micromachines-15-01205-f005:**
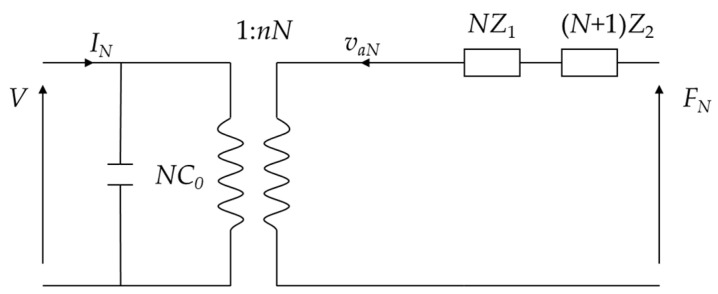
Equivalent circuit of the piezoelectric stack.

**Figure 6 micromachines-15-01205-f006:**
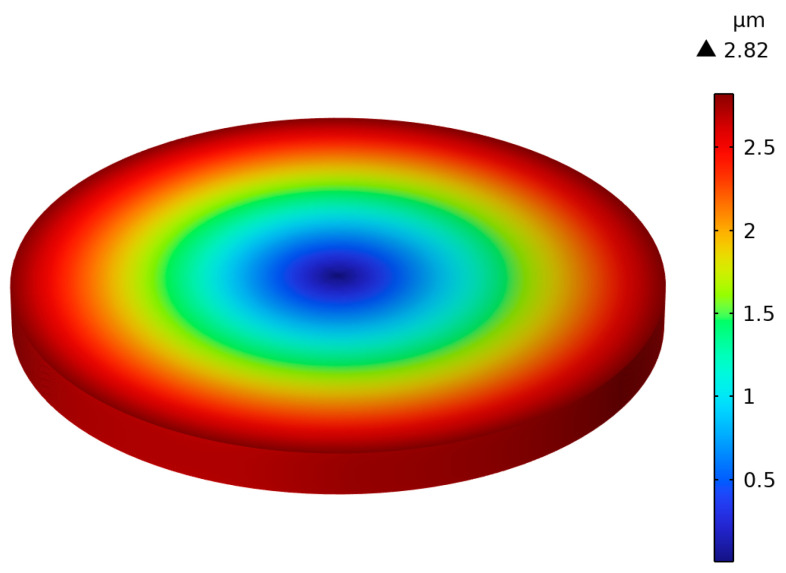
Radial displacement of a piezoelectric stack excited by a voltage of 80 V and 100 Hz.

**Figure 7 micromachines-15-01205-f007:**
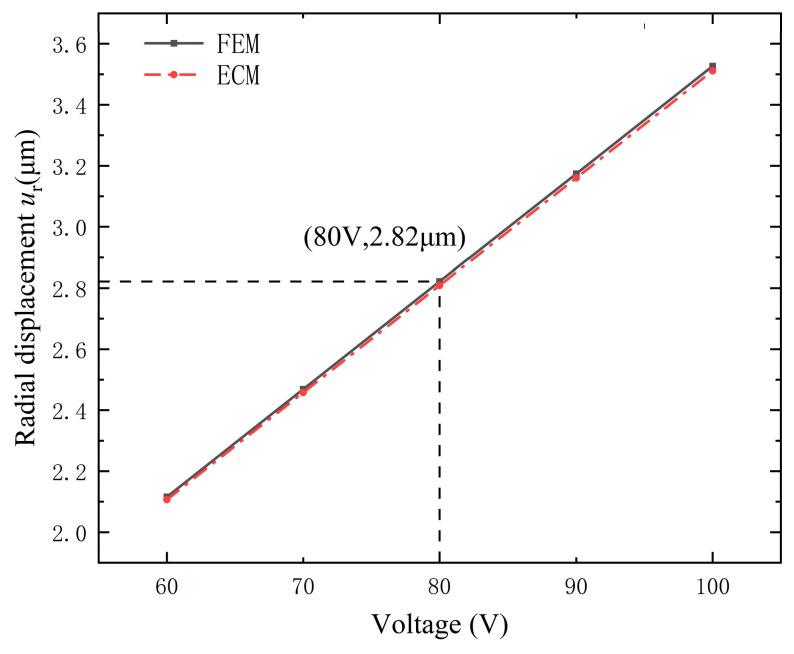
Maximum radial displacement versus driving voltage. FEM: finite element method, ECM: equivalent circuit method.

**Figure 8 micromachines-15-01205-f008:**
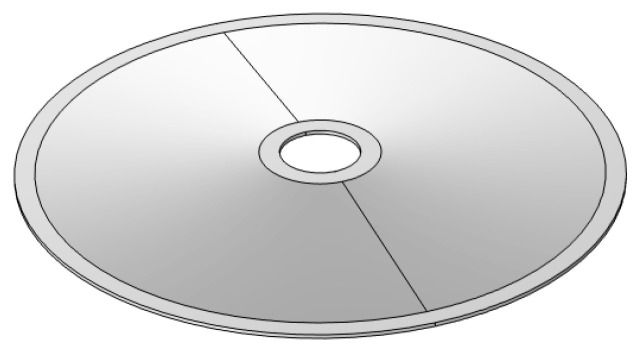
Cymbal disk.

**Figure 9 micromachines-15-01205-f009:**
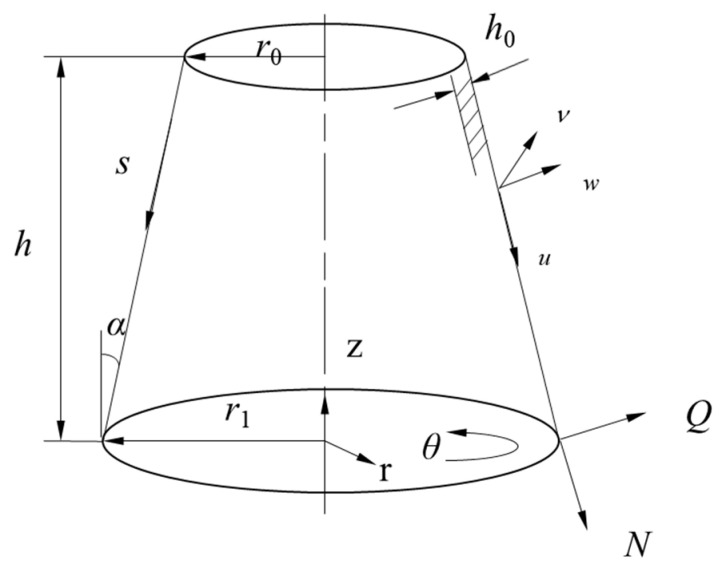
Conical shell model of the cymbal disk.

**Figure 10 micromachines-15-01205-f010:**
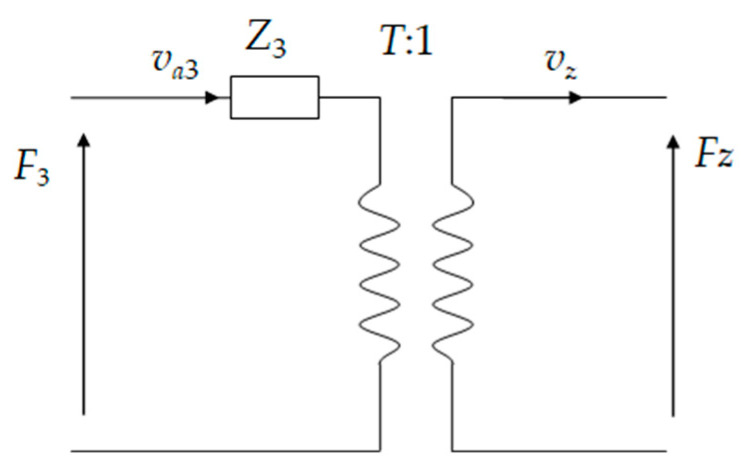
Equivalent circuit of the cymbal structure.

**Figure 11 micromachines-15-01205-f011:**
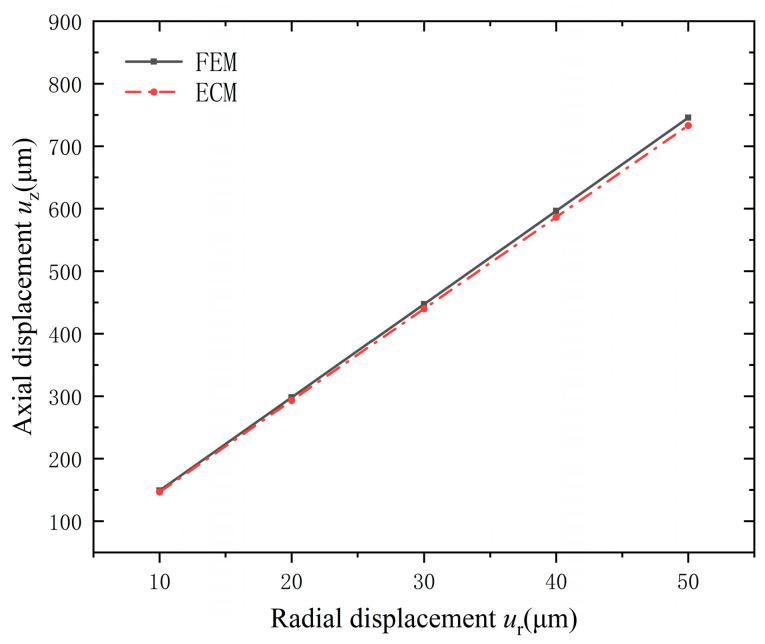
Axial displacement versus radial displacement of the cymbal.

**Figure 12 micromachines-15-01205-f012:**
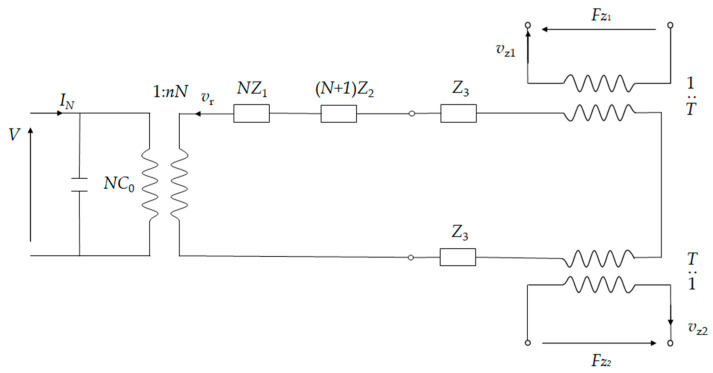
Equivalent circuit of the stacked cymbal vibrator.

**Figure 13 micromachines-15-01205-f013:**
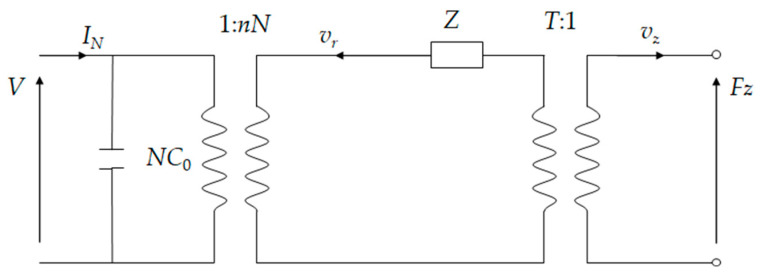
Simplified equivalent circuit of the stacked cymbal vibrator.

**Figure 14 micromachines-15-01205-f014:**
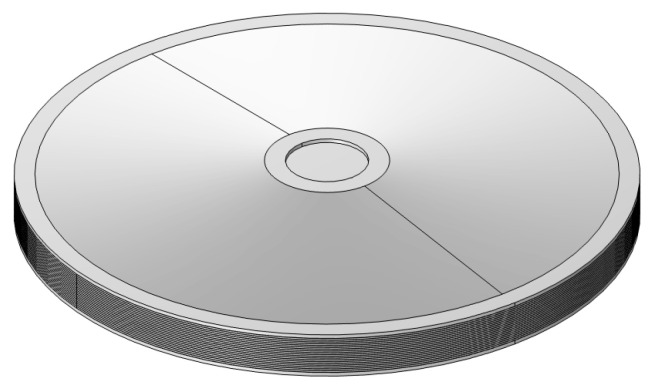
The stacked cymbal vibrator model.

**Figure 15 micromachines-15-01205-f015:**
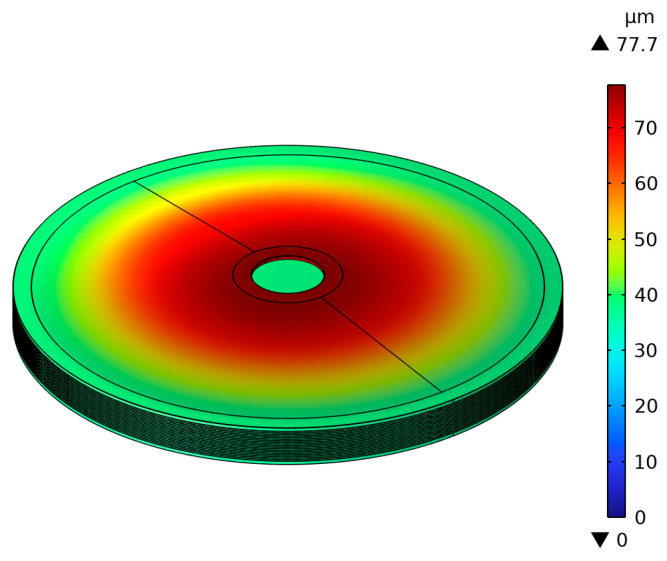
Axial displacement cloud image of the stacked cymbal vibrator model.

**Figure 16 micromachines-15-01205-f016:**
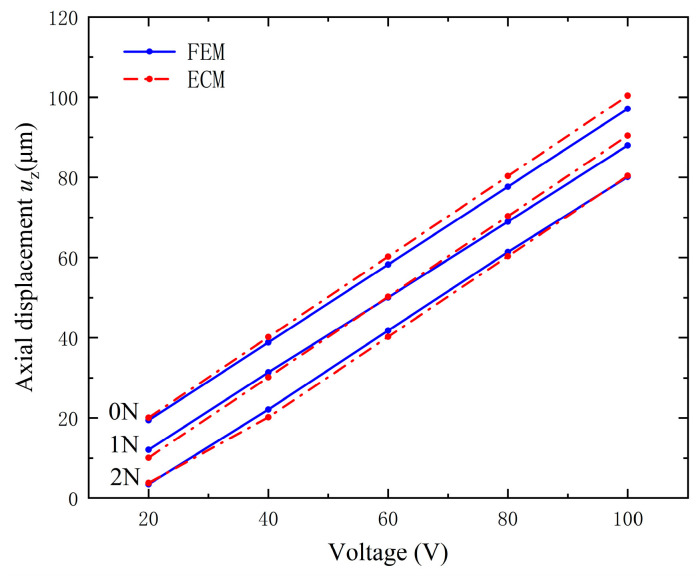
Axial displacement versus operating voltage with different output compression forces, which is calculated by ECM and FEM, respectively.

**Figure 17 micromachines-15-01205-f017:**
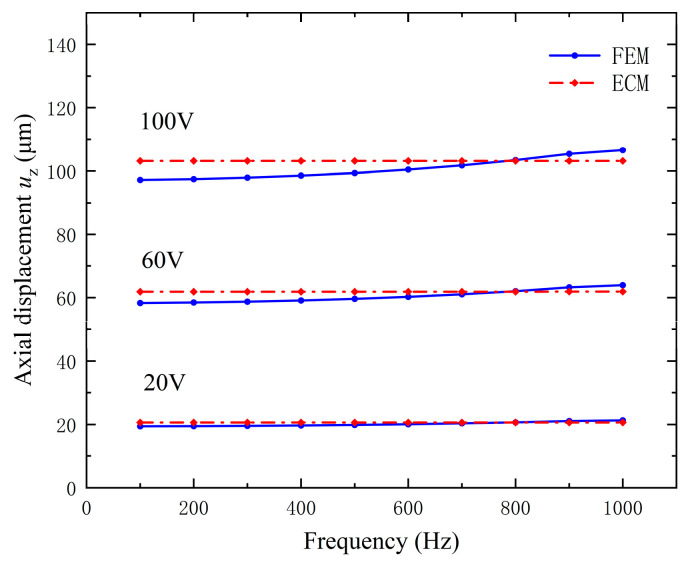
Axial displacement versus frequency under different excitation voltages, which is calculated by ECM and FEM, respectively.

**Table 1 micromachines-15-01205-t001:** Parameters of the piezoelectric stack [30].

Materials	*i*	Density *ρ_i_* (kg/m^3^)	Young’s Modulus *E_i_* (GPa)	Poisson’s Ratio *μ_i_*	Piezoelectric Constants (C/m^2^)			Thickness (μm)
					*e* _31_	*e* _33_	*e* _15_	
PZT-5H	1	7500	\	\	−6.5	23.3	17	45
silver	2	10,490	73	0.38	\	\	\	5

**Table 2 micromachines-15-01205-t002:** Parameters of the cymbal disk [30].

Material	Density *ρ* (kg/m^3^)	Young’s Modulus *E* (GPa)	Poisson’s Ratio *μ*	*α* (°)	*r*_0_ (mm)	*r*_1_ (mm)	*h* (mm)	Thickness (*h*0) (mm)
304SS	7750	193	0.31	86	1.5	7	0.48	0.1

## Data Availability

Data underlying the results presented in this paper are not publicly available at this time, but may be obtained from the authors upon reasonable request.

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
