# Peer review of "Equivalent Circuit of a Stacked Piezoelectric Cymbal Vibrator"

_micromachines, 2024, doi:10.3390/mi15101205_

Round 1
Reviewer 1 Report
Comments and Suggestions for Authors
1. The boundary conditions in Formula 5 are not specified, and the following examples use the four-sided free boundary conditions, while those in Formula 5 May be the four-sided fixed boundary conditions.
2. The picture may be centered
3. The definition and solution process of B1 and B2 in Formula 14 and Formula 5 are not given. Moreover, according to the solution process, whether this model can be understood as an undamped forced vibration model can be calculated according to this method, because the damped forced vibration is closer to the real situation? And the results of displacement and force are more close to the real situation.
4. At the same time, it is noted that only an applied voltage with a frequency of 100Hz is designed in the example. Can we directly conduct frequency domain analysis and design an applied voltage with a frequency of discrete changes?
5. Are the models calculated in Figure 7 and Figure 6 the same model (referring to boundary conditions and structural forms)? If it is the same, the FEM in Figure 7 does not seem to reach the size of 2.87 in Figure 6 at 80V
6. It is said in Figure 17 that ECM has the advantage of high calculation accuracy compared with FEM, but should this conclusion be drawn after comparing the two with the experimental results?
7. Figure 17 shows the harmonic analysis, but does not explore whether the resonance case is out of consideration.
8. Can the relationship between current and charge in Formula 8 give the original expression instead of the direct result?
Comments on the Quality of English LanguageThe quality of full English Language needs to be improved.
Reviewer 2 Report
Comments and Suggestions for Authors
Dear Authors,
My comments related to the manuscript are listed in the attachment.
Kind Regards

Comments on the Quality of English LanguageEnglish should be polished
Round 2
Reviewer 2 Report
Comments and Suggestions for Authors
Dear Authors,
The manuscript can be published after minor revisions (Please increase the quality of some pictures and figures).
Kind regards
Comments on the Quality of English LanguageEnglish is fine
Author Response
Comments 1:
The manuscript can be published after minor revisions (Please increase the quality of some pictures and figures).
Response 1: Thank you. All pictures and figures are checked. The quality of them are enhanced.